# How Can We Improve the Survival of Patients with Colorectal Liver Metastases Using Thermal Ablation?

**DOI:** 10.3390/cancers17020199

**Published:** 2025-01-09

**Authors:** Toshiro Masuda, Toru Beppu, Hirohisa Okabe, Katsunori Imai, Hiromitsu Hayashi

**Affiliations:** 1Department of Surgery, Yamaga City Medical Center, Yamaga 861-0593, Japan; 2Department of Gastroenterological Surgery, Graduate School of Life Sciences, Kumamoto University, Kumamoto 860-8555, Japan

**Keywords:** colorectal liver metastases, propensity score matching, radiofrequency ablation, randomized controlled trial, microwave ablation therapy, liver resection, neoadjuvant chemotherapy, thermal ablation

## Abstract

We can improve the long-term survival of patients with colorectal liver metastases (CRLMs) using thermal ablation. Thermal ablation can provide equivalent long-term survival compared with liver resection for patients with small (≤3 cm) and few CRLMs. We strongly recommend applying thermal ablation for such patients whose liver resection is high risk. Thermal ablation can be combined with liver resection to expand the resectability of numerous bilateral CRLMs. Furthermore, thermal ablation might result in better survival than liver resection in patients receiving effective neoadjuvant chemotherapy. Thermal ablation should be added to systemic chemotherapy for unresectable CRLMs. Microwave ablation therapy, especially microwave thermosphere ablation, can provide better local control and increased disease-free survival than radiofrequency ablation. Furthermore, randomized controlled trials or propensity score matching studies using a prospective database are needed to confirm thermal ablation’s effectiveness and identify the target population that will benefit most from thermal ablation.

## 1. Introduction

Thermal ablation (TA) including radiofrequency ablation (RFA) and microwave ablation therapy (MWA) is widely used for patients with small-size colorectal liver metastases (CRLMs) [1,2,3,4,5,6,7,8,9,10,11]. In its early use, TA was mainly applied in unresectable CRLMs; however, the indication has been expanding to resectable cases, mainly in Western countries [1,2,3,4,5,6,7,8,9,10]. A pivotal randomized controlled trial (RCT) comparing liver resection (LR) and TA was presented during the American Society of Clinical Oncology (ASCO) 2024 meeting [12,13,14]. TA showed noninferiority to LR for CRLMs with a limited number and size. Neoadjuvant chemotherapy (NAC) followed by TA can be used as first-line treatment for small and oligometastatic CRLMs [15,16].

Combining TA with LR can increase resectability, while perioperative chemotherapy further increases the curability [11,17,18,19,20,21,22]. CRLMs are prone to recurrence, which necessitates frequent treatment. TA is beneficial for recurrent CRLMs, which can be diagnosed in small-size and -number tumors by close follow-up [15,23].

In this commentary, we summarize the ways that survival in CRLMs can be improved using TA based on the latest data.

## 2. Thermal Ablation Versus Liver Resection for Colorectal Liver Metastases

Many published papers have compared the utility of LR and TA; however, TA has been indicated mainly for unresectable CRLMs [1,2,3]. Therefore, the background characteristics including tumor and patient factors differed between the two treatment groups. Meta-analyses of studies comparing LR and TA have reported the superiority of LR without performing matching baseline characteristics.

A systematic review was conducted on articles retrieved from the Medline, Embase, and The Cochrane Library databases for studies comparing RFA with LR for CRLMs [3]. RFA had significantly lower complication rates compared with LR (odds ratio [OR] = 0.44, 95% confidence interval [CI] = 0.26–0.75, *p* = 0.002). However, RFA showed higher recurrence rates, specifically, any recurrence (OR = 1.66, 95% CI = 1.15–2.40, *p* = 0.007), local recurrence (OR = 9.56, 95% CI = 6.85–13.35, *p* = 0.001), intrahepatic recurrence (OR = 1.96, 95% CI = 1.34–2.87, *p* = 0.001), and extrahepatic recurrence (OR = 1.21, 95% CI = 0.90–1.63, *p* = 0.22). Furthermore, the 5-year disease-free survival (DFS; hazard ratio [HR] = 2.20, 95% CI = 1.28–3.79, *p* = 0.005) and overall survival (OS; HR = 2.35, 95% CI = 1.49–3.69, *p* = 0.001) were significantly lower in patients treated with RFA.

A recent meta-analysis of 22 studies comparing LR and RFA was reported that included 4385 patients with CRLMs [8]. The patients undergoing RFA showed significantly higher marginal and intrahepatic recurrence rates than LR, with a pooled OR of 7.09 (95% CI:4.56–11.2) and 2.02 (95% CI: 1.24–3.28), respectively. In addition, RFA showed lower 5-year OS and DFS rates than LR, with a pooled OR of 0.60 (95% CI: 0.48–0.74) and 0.74 (95% CI: 0.56–0.97), respectively. No significant difference was observed in the 30-day mortality between RFA and LR.

Subsequently, we developed clinical practice guidelines for the management of liver metastases from extrahepatic primary cancers in 2021 [24]. We then explored the differences between LR and TA in three categories: (1) multiple CRLMs, (2) solitary metastasis, and (3) solitary metastasis and tumors ≤3 cm in diameter. In multiple CRLMs, the 5-year OS, 3-year DFS, and local recurrence rate were significantly better after LR than TA, and the overall complication rates were comparable between the two treatments. Similarly, in a single CRLM, the 5-year OS and local recurrence rates after LR were significantly better than TA, whereas the 3-year DFS and overall complication rates were comparable between the two treatments. Similarly, for single metastasis with tumors ≤ 3 cm in diameter, no differences were observed in the 5-year OS and 3-year DFS between LR and TA. However, the local recurrence rate was significantly greater after TA than after LR. Two studies on a single metastasis ≤ 2 cm in diameter showed that the local recurrence rates were comparable between the two treatments. TA may be a good treatment option for solitary metastasis ≤ 2 cm in diameter; however, TA is not more highly recommended than LR for patients with a CRLM, but is recommended with only a weak level of confidence in the 2021 guidelines. As mentioned earlier, the results of studies that did not match the background factors did not recommend the use of TA for unresectable CRLMs.

To minimize the biases in the background factors in TA and LR, propensity-score matching (PSM) was applied (Table 1). A PSM study using Swedish nationwide registry enrolled patients undergoing MWA or LR as first-line treatment for CRLMs with tumors ≤ 3 cm in diameter between 2013 and 2016 [6]. Therapeutic effects were assessed, adjusting for patient and tumor factors, treatment period, and previous chemotherapy. Before PSM, 82 MWA and 645 LR cases differed significantly in 3-year OS (76% and 69%, respectively; *p* = 0.005). After PSM, 70 MWA and 201 LR cases showed comparable 3-year OS (76% and 76%, *p* = 0.253). Before PSM, patients in the MWA group were significantly older (median age 78.5 years vs. 69 years, *p* = 0.03), with higher American Society of Anesthesiology (ASA) class (*p* < 0.001), worse World Health Organization performance status (*p* = 0.017), and more patients with a Charlson comorbidity index > 11 (*p* = 0.005). Such biases resulted in worse OS in the MWA patients.

Another PSM study analyzed the records retrieved from a multicenter database of patients curatively treated by local therapy between 2000 and 2018 [25]. Patients receiving two-stage therapy, concurrent therapy with the primary tumor, or a combination of multiple therapies were excluded. PSM was applied to minimize the influence of known covariates (i.e., age, ASA, Fong clinical risk score (CRS), location, and T stage of the primary tumor). Before matching, the RFA group included 39 patients, whereas the surgery group included 982 patients. After matching, both groups contained 36 patients. Furthermore, the median (interquartile range, IQR) tumor size and number was 2.5 cm (IQR, 0.8–6.5) and 2 (IQR, 1–5) for RFA, respectively, and 3.4 cm (IQR, 1–7.5) and 1 (IQR, 1–5) for LR, respectively. The RFA and LR groups showed similar overall complication rates according to the Clavien–Dindo classification (17% vs. 33%, respectively; *p* = 0.18) without significant differences in the 5-year DFS and OS (25% vs.37%, respectively; *p* = 0.09; and 42% vs. 53%, respectively; *p* = 0.09). The median day of hospital admission (25th–75th percentile) was significantly smaller in the TA group than in the LR (4.5 [2,3,4,5,6,7] vs. 7 [6,7,8,9]; *p* < 0.01]. After PSM, TA had fewer complications than LR and similar oncologic results. Thus, TA is potentially an appropriate and safe alternative for patients for whom LR is not optimal.

The following papers summarized the results of patients with CRLMs amenable to ablation and resection. Therefore, differences in the oncologic factors were minimal between the LR and TA groups. A prospective multicenter trial (MAVERRIC) was conducted comparing stereotactic MWA (SMWA) and LR for resectable CRLMs [26]. Patients with five or fewer CRLMs with tumors ≤ 3 cm in diameter who were deemed eligible for SMWA and LR during a local multidisciplinary team meeting were intentionally treated with SMWA (study group). The control group was selected from a prospectively maintained Swedish national database and consisted of patients with five or fewer CRLMs with tumors ≤ 3 cm in diameter treated with LR. All patients in the study group (*n* = 98) were matched with the 158 patients in the control group. After PSM, the estimated 3-year OS (78% vs. 76%, respectively) and 5-year OS (56% vs. 58%, respectively) were almost completely equivalent after SMWA and LR. The overall and major complication rates were significantly lower after SMWA (67% vs. 80% reductions, respectively; *p* < 0.01); however, repeated local therapy for intrahepatic lesions was more common after SMWA (increase rate 78%, *p* < 0.01).

A quasi-randomized trial was conducted for CRLMs with tumors ≤3 cm in diameter. The MWA group (*n* = 52) and LR group (*n* = 53) showed similar baseline patient characteristics [27] The mean ± standard deviation overall healthcare-related costs 2 years after the decision to treat were significantly lower in the MWA group than in the LR group (USD 800,964 ± USD 590,182 vs. USD 1,100,059 ± USD 590,671, respectively; *p* < 0.01). The 5-year OS was comparable (50% vs. 54%, respectively; *p* = 0.95). However, MWA was associated with lower morbidity rates, time spent in healthcare facilities, and healthcare-related costs within 2 years of initial treatment, with equivalent long-term survival.

More recently, the results of the Collision trial (multicenter, phase III RCT by the Dutch Colorectal Cancer Group) were presented at the ASCO 2024 meeting [14]. This study investigated the potential noninferiority of TA to LR for patients with small-size CRLMs. The main eligibility criteria were age ≥ 18 years, Eastern Cooperative Oncology Group Performance Status score 0–2, at least one resectable and ablatable CRLM with tumors ≤ 3 cm in diameter, other CRLMs with tumors of any size that were resectable or ≤3 cm that could be ablated, ≤ 10 CRLM, no extrahepatic metastases, and no history of resection or ablation for CRLMs. Patients were stratified into low, intermediate, and high tumor burden subgroups and randomly assigned (1:1) to undergo LR or TA. The primary outcome was OS. Each of the 148 patients with CRLMs was assigned to TA or LR. No biases in patient background were found between the two groups. The median CRLM number was 2, and the mean CRLM size was 14 mm. In the LR group, 35.1% underwent TA, while 18.2% of the TA group underwent LR. No differences were observed in OS (HR = 1.051; 95% CI, 0.695–1.590, *p* = 0.813), local PFS (HR = 0.817, 95% CI: 0.435–1.4543, *p* = 0.530), and distant PFS (HR 1.030, 95% CI: 0.776–1.368, *p* = 0.836). The frequency of adverse events was significantly lower in TA (*p* = 0.001): (overall grade: 46% vs. 19%, respectively; ≥Grade III: 20% vs. 7%, respectively). Procedure-related mortality was 2.1% for LR vs. 0% for TA. The trial was stopped halfway because it met the predefined stopping rules. After a median follow-up time of 28.8 months, no difference in OS was found, with a conditional probability of 90%, thereby proving the hypothesis of noninferiority.

According to studies on resectable and ablatable CRLMs, TA is promising as the standard procedure for patients with small (≤3 cm) and few CRLMs because TA can reduce complications and shorten hospital stays without compromising long-term survival. However, TA is contraindicated for CRLMs with bulky or numerous tumors, tumors adjacent to or invading the Glissonean capsule or blood vessels, and vascular tumor thrombus. In contrast, LR is not suitable for patients with older age, poor PS, poor liver function, and high comorbidity. Thus, TA is an excellent tool because it is less invasive but curative in treating small-size and few CRLMs. Nevertheless, the initial treatment choice for resectable and ablatable cases will be debated.

## 3. Combining Thermal Ablation and Liver Resection to Increase the Resectability of Initially Unresectable Colorectal Liver Metastases

Parenchymal-sparing hepatectomy is the first-choice treatment for multiple CRLMs [28]. TA can be used as an adjunct to LR as a parenchymal-sparing treatment strategy in patients involving deeper lesions [29]. CRLMs that are unresectable using standard procedures (i.e., numerous bilateral CRLMs) are treated with LR using special techniques including portal vein embolization followed by LR, two-stage LR, or associating liver partition and portal vein ligation for staged hepatectomy [30]. Combining TA in both standard and special LR can increase tumor resectability. The essential issue is whether the long-term prognosis of TA + LR is comparable to that of LR alone. 

Based on the clinical practice guidelines for liver metastases, we performed a meta-analysis comparing LR alone and LR + TA for multiple CRLMs [24]. The 3-year OS was significantly greater after LR than after LR + TA, but no differences were observed in the 3-year DFS and overall complication rate. Several other studies and meta-analyses comparing the efficacies of LR alone and LR + TA have been published [11,17,18,19,31,32,33,34]. As some biases in the background factors could have existed in the two groups, definitive results could not be obtained. Specifically for multiple CRLMs ≥ 4 or ≥5, the combination of LR + TA showed a similar long-term survival with LR.

Studies in which the background factors were aligned are discussed subsequently (Table 2). Our colleagues published a PSM study comparing LR + RFA alone vs. LR for multiple CRLMs [18]. Five hundred and fifty-three patients were divided into the LR + RFA (*n* = 31) and LR alone (*n* = 93) groups. Background factors were well balanced in the matched cohort: multilobar CRLMs, tumor number, and tumor diameter were 77% and 77%, 5 (2–25) and 5 (1–33), and 30 mm (10–90 mm) and 30 mm (8–160 mm) in the LR + RFA and LR-alone groups, respectively. The OS and DFS were comparable between the LR + RFA and LR-alone groups. The 5-year OS was 57% and 61% (*p* = 0.649), respectively, while the 5-year DFS was 19% and 17% (*p* = 0.865), respectively. Local recurrence at the ablation site occurred in 4/31 patients (13%); however, no difference was observed in intrahepatic DFS (*p* = 0.705).

The short- and long-term outcomes of LR + intraoperative RFA for patients with multiple initially unresectable CRLMs were investigated using PSM [35]. The study included 67 patients who underwent combined therapy and 268 who underwent LR alone. After PSM, 42 patient pairs were selected. Patients in the LR + intraoperative RFA group had a median number of 10 CRLMs. The operation time was significantly longer in the combination group (276.93 ± 83.30 min and 222.07 ± 72.28 min, respectively; *p* = 0.002). Postoperative overall and major complications were similar between the groups (*p* = 0.362 and *p* =1.000, respectively). The OS, PFS, and hepatic recurrence-free survival (HRFS) were also similar between the two groups (3-year OS: 54.2% and 60.9%, respectively; *p* = 0.389; 3-year PFS: 7.9% and 19.6%, respectively; *p* = 0.148; 3-year HRFS: 16.7% and 31.5%, respectively; *p* = 0.202).

A case-matched analysis comparing LR + MWA and LR was reported [36]. Using a case-matching method based on age, sex, ASA, body mass index, and the number and maximum size of the CRLMs, 20 patients each were selected from the LR + intraoperative MWA or LR-only groups. At the median follow-up of 22.4 ± 17.8 months, 60% in the combination group and 65% in the LR-only group experienced intrahepatic recurrence (*p* = 0.774). No patients in either group experienced recurrence at the same resection or ablation site. No differences were reported between the two groups in RFS (*p* = 0.685) and postresection OS (*p* = 0.151). The combination of intraoperative MWA on LR was not a significant predictive factor for RFS and OS.

More recently, a large-scale nationwide population-based PSM study of trends and OS after combined LR and TA (combination group) was published by a Netherlands study group [37]. Hospital variation in the use of the combination group versus LR alone (LR-alone group) in patients with 2–3 CRLMs with tumors ≤ 3 cm in diameter was assessed. This study included 3593 patients from 2014 to 2022. The combination group increased from 31.7% in 2014 to 47.9% in 2022. Significant hospital variation (range, 5.9–53.8%) was observed in the frequency of the combination group. PSM resulted in 1005 patients in each group. The major complication rates did not differ between the two groups (11.6% vs. 5%, *p* = 1.00). Liver failure occurred less frequently in the combination group (1.9% vs. 0.6%, respectively; *p* = 0.017), and the 5-year OS was comparable (39.3% vs. 33.9%, *p* = 0.145). However, a serious limitation point of this study was the exclusion of parenchymal-sparing hepatectomy.

Four matching studies demonstrated similar results between the LR + TA and LR-alone groups in the long-term survival and recurrence rates including recurrences at the therapeutic sites without an increase in postoperative complications. The choice between the two approaches may mainly depend on the location, number, and size of the CRLMs [36].

## 4. Chemotherapy in Combination with Thermal Ablation for Patients with Colorectal Liver Metastases

### 4.1. Neoadjuvant Chemotherapy Combined with Thermal Ablation

Current systemic chemotherapy with cytotoxic agents and targeted drugs can induce excellent tumor regression and necrosis, along with a high pathological response [38,39]. For patients with high recurrence risk, neoadjuvant chemotherapy before hepatectomy is recommended and enhances RFS [21,40]. These are the backgrounds for attempting NAC before TA. In the period when NAC was not considered, patients who underwent TA had a significantly higher local recurrence rate and worse long-term prognosis than those who underwent LR [1,2].

An attractive PSM study recently compared RFA and hepatectomy for patients with CRLM following NAC in cancers [16] (Table 1). In total, 190 patients with initially resectable CRLMs were enrolled, and all patients received NAC. In the RFA group, the XELOX, FOLFOX, and FOLFIRI regimens were selected 65.6%, 16.4%, and 18.0%, respectively, and were combined with targeted drugs in 21.3% of patients. The median number of NAC cycles in the RFA group was 4 (IQR, 3–6). The baseline characteristics after PSM were identical between the RFA (*n* = 48) and LR (*n* = 48) groups. The RFA group showed significantly lower intraoperative blood transfusion rates (0.0% vs. 8.5%, respectively; *p* = 0.044) and a shorter median hospital stay (2 days vs. 7 days, respectively; *p* < 0.001). Serious complication rates (Clavien–Dindo classification ≥ 3) were lower in the RFA group but not significant (1.6% vs. 10.1%, respectively; *p* = 0.075). Furthermore, the 3-year PFS was superior in the RFA group (55.3% vs. 38.8%, respectively; *p* = 0.035). Multivariate analysis of the overall cohort revealed that the selection of NAC + RFA was an independent predictive factor for good PFS (HR = 1.850; 95% CI, 1.248–2.743; *p* = 0.002). Furthermore, high CRS (3–5; HR, 1.555; 95% CI, 1.087–2.225; *p* = 0.016) and nonresponse to NAC (HR = 1.643; 95% CI, 1.135–2.379; *p* = 0.009) were independent predictors of poor PFS. In contrast, the 3-year OS was comparable between the two groups (73.6% vs. 73.8%, *p* = 0.660).

This is the first report indicating the potential of RFA to provide a better PFS than LR for resectable and ablative CRLMs. Key points included combined NAC and an excellent response to NAC. In contrast, the other matching studies included only 1/4 to 1/2 of patients who received NAC and ablation (Table 1). The maximum diameter of the CRLM tumors before NAC in the RFA group was 3 cm in 33% of patients and > 5 cm in 0%. One key factor is keeping an indication of RFA. The degree of pathological response by NAC may also affect the necrotic effect of RFA. Whether NAC generates unexpected mechanisms that affect RFA needs to be addressed in future studies.

### 4.2. Adjuvant Chemotherapy on Thermal Ablation

A nationwide multicenter observational cohort study for solitary CRLMs compared patients undergoing TA with or without adjuvant chemotherapy [41]. In total, 369 patients with solitary CRLMs treated with TA from October 2010 to May 2023 were investigated. Patient characteristics included tumors smaller than 5 cm, no extrahepatic metastases, and R0 resection for colorectal cancer. Subsequently, 226 and 143 patients were treated with TA + systemic chemotherapy (TAS) and TA, respectively. The median follow-up period was 8.8 years. After PSM, 116 patients were assigned to each of the two groups. The estimated 3-/5-year DFS were 38.3%/28.9% and 34.2%/25.0% in the TAS and TA groups, respectively (HR = 1.12; 95% CI: 0.80–1.56; *p* = 0.52). The estimated 3-/5-year OS rates were 82.2%/70.3% and 73.1%/56.9% in the TAS and TA groups, respectively (HR = 1.60; 95% CI: 0.94–2.72; *p* = 0.08). The subgroup analysis revealed that TAS provided a significantly better OS than TA in high-risk patients with plasma CEA > 5 ng/mL (*p* = 0.036), T stage (III–IV) of primary colon cancer (*p* = 0.034), or Fong’s CRS (1–2) (*p* = 0.041). This study demonstrated that TA can equally and effectively improve the DFS, whether with or without adjuvant chemotherapy. Furthermore, additional adjuvant chemotherapy on TA may improve the OS.

### 4.3. Additional Thermal Ablation on Systemic Chemotherapy

A randomized phase II trial (CLOCC trial) was conducted to compare systemic chemotherapy alone (*n* = 59) and systemic chemotherapy plus RFA (±LR) (*n* = 60) [42]. The primary endpoint was met: a 30-month OS rate of 61.7% (>38%) for the combined treatment group. However, the 30-month OS for chemotherapy was 57.6%, which was higher than the predicted value. The three-year PFS for combined treatment was significantly higher at 27.6% than 10.6% for chemotherapy (HR = 0.63, *p* = 0.025). The median OS was equivalent: 45.3 months for combined treatment and 40.5 months for chemotherapy (*p* = 0.22). Subsequent long-term analysis at a median follow-up of 9.7 years showed that the combined treatment significantly improved the OS. The 3-, 5-, and 8-year OS were 56.9%, 43.1%, and 35.9%, respectively, in the combined treatment group and 55.2%, 30.3%, and 8.9%, respectively, in the chemotherapy group. Based on these results, additional RFA (±LR) is strongly recommended in combination with systemic chemotherapy for surgically nonresectable patients.

Another phase II RCT was conducted to compare the survival benefits in patients with CRLMs undergoing systemic chemotherapy alone or with aggressive local treatment by RFA ± LR [43]. The study included 119 patients with unresectable CRLMs (*n* < 10 and no extrahepatic disease). After a median follow-up of 9.7 years, significantly better OS was observed in the combined modality arm (HR = 0.58; *p* = 0.01). The estimated 3-, 5-, and 8-year OS were 56.9%, 43.1%, and 35.9%, respectively, in the combined modality arm and 55.2%, 30.3%, 8.9%, respectively, in the systemic treatment arm. Combination treatment reduced hepatic progression rates from 78.0% to 46.7%.

## 5. Prognostic Factors for Patients with Colorectal Liver Metastases Treated with Thermal Ablation

Several predictors of outcome were reported for patients with CRLMs after TA. A retrospective analysis of 210 patients with CRLMs undergoing percutaneous MWA showed that multipole tumor (*p* = 0.004; HR: 1.838; CI:1.213–2.784), largest tumor size > 3 cm, (*p* = 0.017; HR: 1.631; CI: 1.093– 2.436), and serum CEA level >30 ng/mL (*p* = 0.032; HR: 1.559; CI: 1.039–2.340) were independent predictors of OS [44]. Two hundred and eighty-four patients with CRLMs treated with ultrasound-guided TA were analyzed [45]. A maximal diameter ≥2.6 cm (HR, 1.59; 95% CI, 1.23 to 2.05), multiple metastases (HR, 1.66; 95% CI, 1.28 to 2.16), and extrahepatic metastases (HR, 1.45; 95% CI, 1.04 to 2.03) were poor prognostic factors, and male sex (HR, 0.75; 95% CI, 0.58 to 0.98) and preoperative chemotherapy (HR, 0.69; 95% CI, 0.52 to 0.92) were good prognostic factors. NAC can provide a better prognosis based on the tumor shrinkage and decrease in CEA levels.

A recent systematic review analyzed the neutrophil-to-lymphocyte (NLR) ratio as a prognostic factor in CRLM patients treated with various local therapies [46]. Two studies included patients treated with RFA. The cutoff values of the NLR for the former and the latter were 5 and 2.5, respectively. Multivariate analysis showed that higher NLR was associated with lower OS (HR = 3.59, 95% CI: 1.54–9.67, *p* = 0.039) and DFS (HR = 3.19, 95% CI: 1.87–8.24, *p* = 0.022,) in the former study, and only increased NLR after surgery was associated with a lower DFS associated with a decrease in DFS (*p* = 0.029). Conversely, it has been reported that the lymphocyte-to-monocyte ratio, and not NLR, can predict survival after RFA for CRLMs [47]. MST was significantly longer in patients with LMR > 3.96% [55 months (95%CI, 37–69)] than in patients with an LMR ≤ 3.96% [34 months (95%CI, 26–39)]. Multivariate analysis showed that the LMR was an independent prognostic marker (HR = 0.53, 0.34–0.85, *p* = 0.007). Max diameter (reference ≤ 30 mm) was another prognostic marker (HR = 0.53, 0.34–0.85, *p* = 0.007).

The Amsterdam Colorectal Liver Met Registry (AmCORE) based study analyzed 520 patients treated with local treatment (LR and/or TA) from 2000 to 2021 [48]. The purpose of this study was to show whether primary tumor laterality, rat sarcoma virus oncogene homolog (RAS), v-RAF mouse sarcoma virus oncogene homolog B (BRAF) mutations, and microsatellite instability (MSI) status could predict the survival outcomes. For RAS mutations, there was no significant difference in OS (*p* = 0.116). In contrast, distant progression-free survival (DPFS) (*p* = 0.001), local tumor progression-free survival (LTPFS) (*p* = 0.039), and local control (LC) (*p* = 0.025) were significantly lower in the RAS mutation group. Although there were no differences in LTPFS between the RAS wildtype and mutated CRLMs following LR (*p* = 0.532), LTPFS was worse for the RAS mutated patients than the RAS wildtype following TA (*p* = 0.037). OS was significantly lower in the BRAF mutation (*p* < 0.001) and MSI (*p* < 0.001) groups following local treatment, while both did not affect the DPFS, LTPFS, and LC. Preoperative knowledge regarding molecular biomarkers may contribute to improved survival outcomes.

## 6. Additional Effects of Thermal Ablation for Patients with Colorectal Liver Metastases

### 6.1. Decrease the Effect of Growth Factor Release Compared to Liver Resection

The experimental study was conducted in a mouse model to determine the effects of RFA on tumor growth and growth factor expression [49]. Baseline hepatocyte growth factor (HGF) and basic fibroblast growth factor (bFGF) expression was significantly higher in the tumor-bearing mice compared to the tumor-free controls. Growth factor expression increased after LR but decreased after RFA; LR but not RFA accelerated tumor growth in the remnant liver.

### 6.2. Immunomodulatory Effect of Thermal Ablation

The TA of liver tumor can produce in situ tumoral antigens and stimulate specific immune responses. The immunomodulatory effects induced by RFA treatment were assessed for patients with liver metastases (LM) and hepatocellular carcinoma (HCC) [50]. After RFA, lymphocyte subsets of CD3+ T cells, especially CD4+, were reduced in the LM patients, while no changes were observed in the HCC patients. In addition, RFA induced the migration of naive and memory CD62L+ T cells from circulation to the tissues. Characterizing T cell function, the proliferative response to phytohemagglutinin was greatly increased in LM patients 48 h after RFA. Furthermore, an increase in circulating B cells was observed only in LM patients. These results indicate that the application of RFA can exert an activating effect on the immune system of LM patients.

Papers including experimental and clinical studies focusing on immune responses following TA of the liver tumor were reviewed [51]. After TA, a cellular response is elicited by dendritic cells presenting antigens to specific CD41 T cells, which in turn stimulate natural killer cells or CD81 cytotoxic cells. The local release of intracellular debris activates Kupffer cells to produce inflammatory cytokines, which in turn activate and react on nearby monocytes/macrophages or specific T cells, leading to a systemic response.

In a clinical situation, the immunomodulatory effects of RFA and LR on liver cancer were compared [52]. In contrast to the increase in the LR group, a decrease in NLR and platelet-to-lymphocyte ratio (PLR) was observed after RFA. A significant decrease and a non-significant increase in CD4+ and CD39+ lymphocytes were observed in the RFA and LR groups, respectively. These results suggest that TA may have an additional beneficial effect on survival.

## 7. Conclusions and Future Perspective

TA can improve the long-term survival for selected patients with CRLMs. TA should be chosen for patients with small (≤3 cm) and few CRLMs who are at high risk for LR, namely, those with older age, poor PS, poor liver function, and high comorbidity. High postoperative complication rates and more significant intraoperative blood loss and blood transfusion can increase postoperative recurrence and worsen the OS in LR for CRLMs [53,54]. In contrast, TA can decrease surgical insults, resulting in better long-term survival than those in LR. Furthermore, when limited to patients receiving NAC, TA may provide better survival than LR [16].

This year, Adam R. and the Collaborative TransMet group published a pivotal RCT paper on liver transplantation (LT) [55]. In the intention-to-treat analysis, 5-year OS was superior at 56.6% (95% CI 43.2–74.1) for LT plus chemotherapy than 12.6% (5.2–30.1) for chemotherapy alone (HR 0.37 [95% CI 0.21–0·65]; *p* = 0.0003). The standing point of TA for LT in this paper is unknown. However, a novel pre-LT protocol [56] was introduced to improve tumor biology and reduce the tumor burden, and also recommended aggressive pretransplant liver-directed treatment including TA.

A recent meta-analysis (MWA: 316 patients; RFA: 332 patients) demonstrated that MWA could significantly decrease local tumor progression (*p* < 0.05) and significantly improve the DFS rates compared to RFA [57].

In particular, microwave thermosphere ablation (MTA) using saline infusion to cool the antenna could significantly decrease the total ablation time, and the selection of MTA was the independent predictive factor for excellent local recurrence (HR 0.39, *p* = 0.015) [58]. Evidence levels in this field are not very high; therefore, the selection of MWA and RFA is an ongoing topic for consideration.

TA can be combined with LR to expand the resectability of numerous bilateral CRLMs that primarily involve multiple deep lesions. For initially unresectable CRLMs, conversion LR with or without TA after systemic chemotherapy can improve the OS compared with systemic chemotherapy alone [39]. Even for continuously unresectable CRLMs, additional TA on systemic chemotherapy results in significant survival benefits [42,43].

Finally, RCT or PSM studies using an extensive prospective database are required to confirm the effectiveness of TA and to determine the target populations that will benefit most from TA in the future.

## Figures and Tables

**Table 1 cancers-17-00199-t001:** Comparison of thermal ablation versus liver resection using matching baseline characteristics.

REF Number	Publish Year	Matching Method	Size and Number	Patient Number	Treatment Method	3-Year PFS (%)	5-Year DFS (%)	3-Year OS (%)	5-Year OS (%)	Major Complication Rates (%)	Hospital Day	Patients Received NAC (%)
6	2020	PSM	≤3 cm	70	MWA			75.5		7 ^#^		50.0
				201	LR			76.3		16.4 ^#^		52.2
16	2022	PSM	≤5 cm	48	RFA	55.3 ^#^		73.6		1.6	2 ^#3^	100
				48	LR	28.8 ^#^		73.8		10.1	7 ^#3^	100
25	2022	PSM	Median: 2.5 cm and 2	36	RFA		25		42	ALL: 17	4.5 ^##^	42
				36	LR		37		53	ALL: 33	7 ^##^	33
26	2023	PSM	≤3 cm and ≤5	98	SMWA				56	2 ^##^		33
				158	LR				58	10 ^##^		37
27	2023	quasi-RCT	≤3 cm	52	MWA				50	3.8 ^#3^	Post 1 ^#3^	26.1
				53	LR				54	26.4 ^#3^	Post 7 ^#3^	41.5
14	2024	RCT	≤3 cm and ≤10	148	TA			70 ^$^	52 ^$^	7 ^#3^	1 ^#3^	24.3
				148	LR			70 ^$^	58 ^$^	20 ^#3^	4 ^#3^	20.3

REF, reference; PFS, progression-free survival; DFS, disease-free survival; OS, overall survival; Major complications, complications ≧ Clavien–Dindo grade III; NAC, neoadjuvant chemotherapy; PSM, propensity-score matching; LR, liver resection; TA, thermal ablation; MMA, microwave ablation; RFA, radiofrequency ablation; SMWA, stereotactic microwave ablation; ALL, all complications; Post, postoperative hospital day; RCT, randomized controlled trial; #, *p* < 0.05; ##, *p* < 0.01; #3: *p* < 0.001; $: estimated value.

**Table 2 cancers-17-00199-t002:** Comparison of liver resection versus liver resection plus thermal ablation using matching baseline characteristics.

REF Number	Publish Year	Matching Method	Size and Number	Patient Number	Treatment Method	3-Year PFS (%)	5-Year DFS (%)	3-Year OS (%)	5-Year OS (%)	Major Complication Rates (%)	Hospital Day	Patients Received NAC (%)
18	2017	PSM (1:3)	30 (10–90) mm 5 (2–25)	31	LR + RFA		19.4		57.3	23		97
				93	LR		17.3		61.1	24		95
35	2021	PSM	≦30 mm, ≧2	42	LR + IORFA	7.9		54.2		4.8		97.6
				42	LR	19.6		60.9		4.8		97.6
36	2022	case-matched analysis	High tumor burden 25% in each	20	LR + IMWA		mRFS 9.5		MST 53.0	5	9.8	75
				20	LR		2.4		32.5	15	13.7	70
37	2024	PSM	2–3 CRLMs and 3 cm	1005	LR + TA				33.9	11.5		46.3
				1005	LR				39.3	11.6		44

REF, reference; PFS, progression-free survival; DFS, disease-free survival; OS, overall survival; Major complications, complications ≧ Clavien–Dindo grade III; NAC, neoadjuvant chemotherapy; PSM, propensity-score matching; LR, liver resection; RFA, radiofrequency ablation; IORFA, intraoperative RFA; IMMA, intraoperative microwave ablation; TA, thermal ablation; mRFS, median recurrence-free survival; MST, median survival time.

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
