# Peer review of "How Can We Improve the Survival of Patients with Colorectal Liver Metastases Using Thermal Ablation?"

_cancers, 2025, doi:10.3390/cancers17020199_

Round 1

Reviewer 1 Report

Comments and Suggestions for Authors

Dear authors

I read with great interest the manuscript "How Can We Improve the Survival of Patients with Colorectal Liver Metastases Using Thermal Ablation? ". I would like to congratulate the authors for the relevance of the study and for the expression of the results. I have however few suggestions before acceptance:

- An English brush up is required

- In the section "Neoadjuvant chemotherapy combined with thermal ablation ", a relevant study should be added. Indeed, Benhaim et al. in HPB 2018 (DOI: 10.1016/j.hpb.2017.08.023) reported that for RFA ablation the size have to be assessed before starting chemotherapy and that RFA is not the optimal treatment for CRLM > 25 mm at baseline evaluation.

- In the section, "Conclusions and future perspective", The possibility of associating Pringle maneuver to thermal ablation (DOI: 10.1007/s00268-020-05379-4 ) is also a promising perspective to reduce the heat sink effect and to treat more lesions next to large vessels with better local control 

Comments on the Quality of English Language

English revision is required

Author Response

Table 2 was newly made and uploaded.

Thank you very much.

Reviewer 2 Report

Comments and Suggestions for Authors

The paper is interesting and well written. The authors should add a paragraph on the prognostic factors in these patients, as most of the benefit could be obtained if we are able to identify those patients who would benefit the most from the treatment. In this regard, cite the recent paper (PMID: 27122671)

The authors should comment more also on the potential immunomodulatory effect of thermal ablation in these patients

The authors might be aware a recent landmarck paper has been published in the Lancet about the impact of OLT in (selected) patients with liver metastases from CRC. How the ablative treatments could be of help in this new scenario? (downstaging? bridging to OLT?)

Some figues could improve the quality of the paper

Author Response

#1 The paper is interesting and well written. The authors should add a
paragraph on the prognostic factors in these patients, as most of the
benefit could be obtained if we are able to identify those patients who
would benefit the most from the treatment. In this regard, cite the
recent paper (PMID: 27122671)

Thank you for your valuable comment.

We have created a new paragraph 5 about the prognostic factors for CRLM patients treated with TA and cited some papers, including PMID: 27122671.

  1. Prognostic factors for patients with colorectal liver metastases treated with thermal ablation

Several predictors of outcome were reported for patients with CRLM after TA. A retrospective analysis of 210 patients with CRLM undergoing percutaneous MWA showed that multipole tumor (P = 0.004; HR: 1.838; CI:1.213– 2.784), largest tumor size >3 cm, (P = 0.017; HR: 1.631; CI: 1.093– 2.436), and serum CEA level >30 ng/ml (P = 0.032; HR: 1.559; CI: 1.039–2.340) were independent predictors of OS [44].Two hundred eighty-four patients with CRLM treated with ultrasound-guided TA were analyzed [45]. A maximal diameter ≥2.6 cm (HR, 1.59; 95% CI, 1.23 to 2.05), multiple metastases (HR, 1.66; 95% CI, 1.28 to 2.16), and extrahepatic metastases (HR, 1.45; 95% CI, 1.04 to 2.03) were poor prognostic factors, and male sex (HR, 0.75; 95% CI, 0.58 to 0.98) and preoperative chemotherapy (HR, 0.69; 95% CI, 0.52 to 0.92) were good prognostic factors. NAC can provide a better prognosis based on the tumor shrinkage and decrease in CEA levels.

A recent systematic review analyzed the neutrophil-to-lymphocyte (NLR) ratio as a prognostic factor in CRLM patients treated with various local therapies [46]. Two studies included patients treated with RFA. The cutoff values of NLR for the former and the latter were 5 and 2.5, respectively. Multivariate analysis showed that higher NLR was associated with lower OS (HR = 3.59, 95% CI: 1.54-9.67, P =0.039) and DFS (HR = 3.19, 95% CI: 1.87-8.24, P =0.022,) in the former study and only increased NLR after surgery was associated with lower DFS associated with a decrease in DFS (P = 0.029). Conversely, it has been reported that lymphocyte-to-monocyte ratio, not NLR, can predict survival after RFA for CRLM [47]. MST was significantly longer in patients with LMR > 3.96% [55 months (95%CI, 37-69)] than in patients with LMR ≤ 3.96% [34 months (95%CI, 26-39)]. Multivariate analysis showed that LMR was an independent prognostic marker (HR = 0.53, 0.34-0.85, P = 0.007). Max diameter (reference ≤ 30 mm) was another prognostic marker (HR = 0.53, 0.34-0.85, P = 0.007).

The Amsterdam Colorectal Liver Met Registry (AmCORE) based study analyzed 520 patients treated with local treatment (LR and/or TA) from 2000 to 2021 [48]. The purpose of this study was to show whether primary tumor laterality, rat sarcoma virus oncogene homolog (RAS), v-RAF mouse sarcoma virus oncogene homolog B (BRAF) mutations, and microsatellite instability (MSI) status could predict survival outcomes. For RAS mutations, there was no significant difference in OS (P = 0.116). Still, in contrast, distant progression-free survival (DPFS) (P = 0.001), local tumor progression-free survival (LTPFS) (P = 0.039), and local control (LC) (P = 0.025) were significantly lower in the RAS mutation group. Although there were no differences in LTPFS between RAS wildtype and mutated CRLM following LR (P = 0.532), LTPFS was worse for RAS mutated patients than RAS wildtype following TA (P = 0.037). OS was significantly lower in the BRAF mutation (P < 0.001) and MSI (P < 0.001) groups following local treatment, while both did not affect DPFS, LTPFS, and LC. Preoperative knowledge regarding molecular biomarkers may contribute to improved survival outcomes.

#2 The authors should comment more also on the potential immunomodulatory
effect of thermal ablation in these patients

Thank you for your informative comment.

We have added a new paragraph 6 about the additional effects, including the potential immunomodulatory effect of thermal ablation on CRLM.

  1. Additional effects of thermal ablation for patients with colorectal liver metastases

6.1. Decrease the effect of growth factor release compared to liver resection

The experimental study was conducted in a mouse model to determine the effects of RFA on tumor growth and growth factor expression [49]. Baseline hepatocyte growth factor (HGF) and basic fibroblast growth factor (bFGF) expression was significantly higher in tumor-bearing mice compared to tumor-free controls. Growth factor expression increased after LR but decreased after RFA; LR but not RFA accelerated tumor growth in the remnant liver.

6.2. Immunomodulatory effect of thermal ablation

TA of liver tumor can produce in situ tumoral antigens and stimulate specific immune responses. The immunomodulatory effects induced by RFA treatment were assessed for patients with liver metastases (LM) and hepatocellular carcinoma (HCC) [50]. After RFA, lymphocyte subsets of CD3+ T cells, especially CD4+, were reduced in LM patients, while no changes were observed in HCC patients. In addition, RFA induced migration of naive and memory CD62L+ T cells from circulation to tissues. Characterizing T cell function, the proliferative response to phytohemagglutinin was greatly increased in LM patients 48 hours after RFA. Furthermore, an increase in circulating B cells was observed only in LM patients. These results indicate that the application of RFA can exert an activating effect on the immune system of LM patients.

Papers, including experimental and clinical studies focusing on immune responses following TA of the liver tumor, were reviewed [51]. After TA, a cellular response is elicited by dendritic cells presenting antigen to specific CD41 T cells, which in turn stimulate natural killer cells or CD81 cytotoxic cells. Local release of intracellular debris activates Kupffer cells to produce inflammatory cytokines, which in turn activate and react on nearby monocytes/macrophages or specific T cells, leading to a systemic response.

In a clinical situation, the immunomodulatory effects of RFA and LR on liver cancer were compared [52]. In contrast to the increase in the LR group, a decrease in NLR and platelet-to-lymphocyte ratio (PLR) was observed after RFA. A significant decrease and a non-significant increase in CD4+ CD39 + lymphocytes were observed in the RFA and LR groups respectively. These results suggest that TA may have an additional beneficial effect on survival.

#3 The authors might be aware a recent landmarck paper has been published
in the Lancet about the impact of OLT in (selected) patients with liver
metastases from CRC. How the ablative treatments could be of help in
this new scenario? (downstaging? bridging to OLT?)

A pivotal paper about liver transplantation and chemotherapy was published this year, so we have added the following paragraph as, “7. Conclusions and future perspective”.

This year, Adam R. and the Collaborative TransMet group published a pivotal RCT paper about liver transplantation (LT) [55]. By the intention-to-treat analysis, 5-year OS was superior 56∙6% (95% CI 43∙2−74∙1) for LT plus chemotherapy than 12∙6% (5∙2–30∙1) for chemotherapy alone (HR 0·37 [95% CI 0·21−0·65]; p=0·0003). The standing point of TA for LT in this paper was unknown. Still, a novel pre-LT protocol [56] was introduced to improve tumor biology and reduce tumor burden and recommended aggressive pretransplant liver-directed treatment, including TA.

#4 Some figures could improve the quality of the paper

We are sorry but we have already added Table 2.

Round 2

Reviewer 2 Report

Comments and Suggestions for Authors

The revised manuscript is OK. Thank you!